# Assessing the fitness of a dual-antiviral drug resistant human influenza virus in the ferret model

Harry L. Stannard [1], Edin J. Mifsud[1], Steffen Wildum [2], Sook Kwan Brown[1], Paulina Koszalka[1], Takao Shishido[3], Satoshi Kojima[3], Shinya Omoto [3], Keiko Baba[3], Klaus Kuhlbusch[2], Aeron C. Hurt[2] & Ian G. Barr [1,4 ✉]

Influenza antivirals are important tools in our fight against annual influenza epidemics and future influenza pandemics. Combinations of antivirals may reduce the likelihood of drug resistance and improve clinical outcomes. Previously, two hospitalised immunocompromised influenza patients, who received a combination of a neuraminidase inhibitor and baloxavir marboxil, shed influenza viruses resistant to both drugs. Here-in, the replicative fitness of one of these A(H1N1)pdm09 virus isolates with dual resistance mutations (NA-H275Y and PA-I38T) was similar to wild type virus (WT) in vitro, but reduced in the upper respiratory tracts of challenged ferrets. The dual-mutant virus transmitted well between ferrets in an airborne transmission model, but was outcompeted by the WT when the two viruses were co-administered. These results indicate the dual-mutant virus had a moderate loss of viral fitness compared to the WT virus, suggesting that while person-to-person transmission of the dual-resistant virus may be possible, widespread community transmission is unlikely.

[1] WHO Collaborating Centre for Reference and Research on Influenza, at the Peter Doherty Institute for Infection and Immunity, Melbourne, VIC, Australia. [2] F. Hoffmann-La Roche Ltd, Basel, Switzerland. [3] Shionogi & Co., Ltd, Osaka, Japan. [4] Department of Microbiology and Immunology, at the Peter Doherty Institute for Infection and Immunity, the University of Melbourne, Melbourne, VIC, Australia. ✉email: ian.barr@influenzacentre.org

Influenza virus infections are a major and continued burden on human health. Influenza seasonal epidemics may cause up to 1 billion infections and 650,000 deaths worldwide each year[1,2]. In combination with seasonal influenza vaccines, antiviral drugs are an important tool in reducing the disease burden of infected patients, particularly in a pandemic setting, where unavoidable manufacturing and testing requirements will delay vaccine development for a novel strain. As a result, many countries maintain antiviral stockpiles as a pandemic contingency[3]. The most recent influenza pandemic occurred in 2009 when a novel influenza A(H1N1) pandemic virus emerged from a swine-origin virus that had undergone several reassortment events, allowing efficient transmission in humans and rapid global spread[4,5].

There are currently five licensed influenza antiviral drugs recommended for clinical use, although not all are available globally. These include four neuraminidase inhibitors (NAIs), oseltamivir (OST), zanamivir (ZAN), laninamivir (LAN) and peramivir (PER), and a single polymerase acidic (PA) endonuclease inhibitor, baloxavir marboxil. Adamantanes (or M2 inhibitors) are a class of licensed antivirals that have been discontinued from clinical use since 2009, because most circulating viruses are resistant due to the S31N amino acid (AA) substitution in the M2 gene[6,7]. In 2008-2009, widespread resistance to OST and PER occurred due to the neuraminidase (NA) substitution H275Y that was identified in 93-100% of influenza A(H1N1) viruses characterised globally[8,9].

Baloxavir marboxil (BXM), a recently developed inhibitor of the PA cap-dependent endonuclease, has been shown to rapidly reduce viral shedding and reduce symptom duration and complications associated with influenza infection[10,11]. However, BXM treatment can select for influenza viruses with reduced susceptibility to this drug, with the most commonly detected resistance mutation being PA-I38T. In clinical trials, post-treatment influenza virus analysis showed that 2-10% of adults and approximately 20% of children shed virus with a PA-I38T substitutions (or other variants at this locus)[10,12–16]. The selection of treatment-emergent NA-H275Y[17,18] or PA-I38T[12,16] viruses has been associated with longer time to symptom alleviation in some patients compared to patients infected with drug susceptible viruses[19]. In the event that drug resistant viruses emerge, switching from OST to BXM treatment has been shown to improve clinical outcomes for infections with OST-resistant NA-H275Y influenza viruses[20–22]. Surveillance data on the antiviral susceptibility of circulating seasonal influenza viruses from patients in the community (most of whom had not been treated with an antiviral) are reported regularly by the WHO global influenza surveillance and response system. In general, the frequency of viruses tested with reduced susceptibility to at least one of the NAIs or BXM in globally circulating influenza viruses has been very low, at <1% of viruses tested in 2018-2020. However, in Japan, where BXM was more routinely used, the incidence of viruses with reduced susceptibility to BXM was 4.5% (41/991) during 2018-2019[23]. More recently influenza viruses collected from February 2021 to February 2022 revealed only two out of 2953 (0.07%) influenza viruses with reduced inhibition by NAIs (A246V and D197N), and only one of 2443 (0.04%) influenza viruses with reduced susceptibility to BXM (PA-E23K)[24,25], however, this was during an unusual period of very low global circulation of influenza viruses as a consequence of the measures taken to mitigate the COVID-19 pandemic.

Combination antiviral drug therapy is the cornerstone for treating chronic infections by viruses such as HIV and Hepatitis C to avoid the generation of resistance[26] and may offer similar advantages in treating severe cases of influenza[27–30]. Combination therapy with OST and BXM in mice was more effective than either monotherapy to prevent the selection of treatment-emergent resistant influenza viruses[29]. A phase III randomised,

double-blind, placebo-controlled, multicentre study was conducted by Hoffmann-La Roche in 2019-2020 (FLAGSTONE; NCT03684044) to assess the efficacy and safety of the standard of care NAI (often OST, depending on individual hospital practises) treatment in combination with BXM (or placebo) in hospitalised adolescents and adults patients (≥12 years) with severe influenza. In this study, A(H1N1)pdm09 viruses with treatment-emergent dual-mutations (NA-H275Y and PA-I38T) were identified in two immunocompromised (undergoing cancer treatment) patients treated with OST or PER in combination with BXM. These viruses were considered to have reduced susceptibility to OST, PER and BXM[31].

The spread of viruses with reduced susceptibility to BXM and OST/PER in the community would severely reduce current treatment options for influenza. However, the fitness and transmissibility of A(H1N1)pdm09 influenza viruses with NA-H275Y and PA-I38T dual substitutions is unknown. This study assessed the replicative capacity, airborne transmissibility and relative fitness of such viruses isolated from patient samples using in vitro assays and the ferret challenge model.

## Results

### In vitro characterisation of virus isolates containing single and double mutant NA-H275Y and PA-I38T viruses.
Original respiratory clinical samples from a 71-year-old South Korean female patient infected with influenza A(H1N1)pdm09, who was participating in the FLAGSTONE study[31], were obtained and sequenced from Day 1 (d1), Day 4 (d4) and Day 10 (d10) of study enrolment, and the proportions of viruses with NA-H275Y, PA-I38T and PA-P325Q were determined as shown in Supplementary Fig. 1. Purified virus clones were obtained by two rounds of plaque purification from these original isolates, and the resulting viruses were then sequenced to confirm that pure isolates of the desired genotype had been obtained; wild type virus (WT), which had NA-H275 and PA-I38, a single mutant with NA-H275Y, a single mutant with PA-I38T, and a double mutant with NA-H275Y + PA-I38T. We also noted three additional mutations in these isolates; PA-P325Q, which occurred in the patient before antiviral treatment, as well as hemagglutinin (HA)-A204T (H1 numbering, beginning from Met) and viral polymerase 1 (PB1)-Y129H mutations that occurred only after passaging the WT virus in cell culture (Supplementary Table 1).

Susceptibility to baloxavir acid (BXA; the active form of BXM) and all four NAIs, oseltamivir carboxylate (OSC; the active form of OST), PER, LAN, and ZAN, were determined by phenotypic assays for all four viruses. Virus isolates with PA-I38T demonstrated highly reduced susceptibility to BXA ($EC_{50}$ fold change of 373 for PA-I38T and 190 for NA-H275Y + PA-I38T; when compared to the $EC_{50}$ of PA-I38 wild type A(H1N1)pdm09 viruses). Virus isolates with NA-H275Y (NA-H275Y and NA-H275Y + PA-I38T) demonstrated highly reduced susceptibility to OSC and PER ($IC_{50}$ fold change of both viruses of approximately 600 to OSC and approximately 100 to PER when compared to the $IC_{50}$ of NA-H275 wild type A(H1N1)pdm09 viruses). There was no change in susceptibility against LAN or ZAN ($IC_{50}$ fold change <3) for any of the mutant viruses. All isolates with a wild type PA (I38) and/or NA (H275), demonstrated sensitivity to BXA or all four NAIs, respectively (Supplementary Table 2).

The replication kinetics of the WT, NA-H275Y single mutant, PA-I38T single mutant and NA-H275Y + PA-I38T double mutant viruses were analysed in vitro. Influenza virus replication at a high multiplicity of infection (MOI) (5) sampled over 12 h and at a low MOI (0.01) over 102 h were performed to assess single- or multi-stage viral replication, respectively. No substantial differences were observed between the replication kinetics

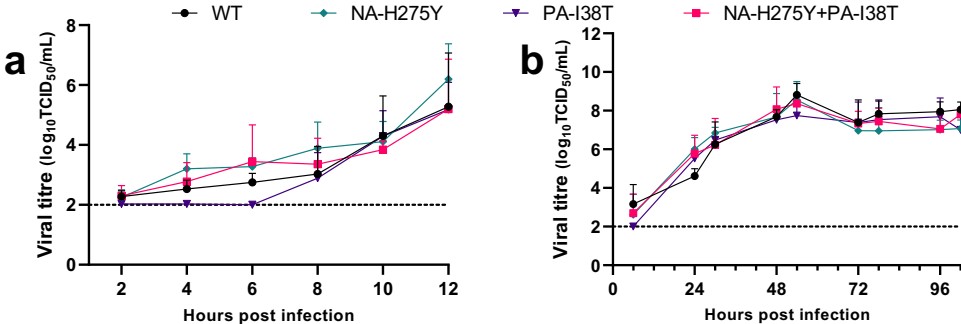

**Fig. 1 Virus growth in single and multi-stage replication cycles.** MDCK cells were infected with virus isolates containing one or both NA-H275Y and PA-I38T mutations. **a** Single-stage replication assay (MOI 5) with sampling every 2 h for a total of 12 h. **b** Multi-stage replication assay (MOI 0.01) and sampled at 6, 24, 30, 48, 54, 72, 78, 96, 102 h. Infectious virus in the supernatant was quantified by $TCID_{50}$. Each symbol represents the mean amount of virus present at each time point and the error bars depict one standard deviation (SD). Data in each graph is the combination of three independently conducted experiments performed in triplicate. Limit of detection (LOD) is shown as a dotted horizontal line, all values below LOD equal two $\log_{10}TCID_{50}/$ mL. No statistically significant differences (adjusted $p < 0.05$) were observed (Two-way ANOVA with Sidak's multiple comparisons).

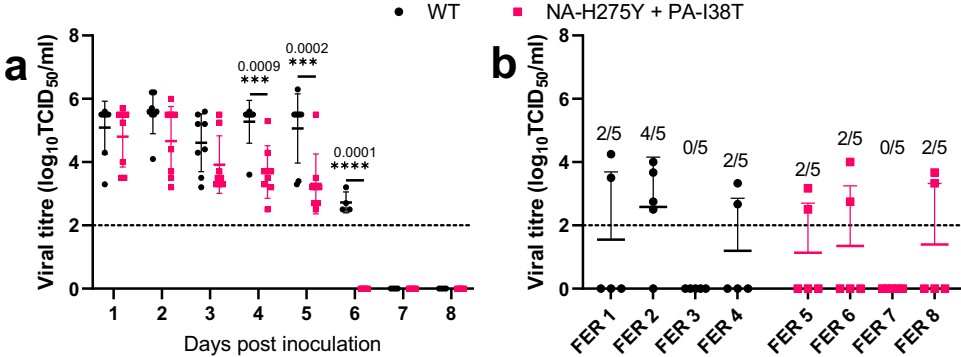

**Fig. 2 Viral shedding and lung lobe viral load of WT and NA-H275Y + PA-I38T virus isolate infected ferrets.** Ferrets ($n = 8$) infected by the intranasal route with 4 $\log_{10}TCID_{50}/$mL (in 500 µL of PBS) with WT (d1 isolate, black) or NA-H275Y + PA-I38T (d10 isolate, red) purified virus isolates were nasally washed daily for 9 days. Half of the ferrets were culled at day five post-inoculation to harvest the five major lung lobes. **a** The viral shedding from the ferret nasal washes was measured by $TCID_{50}$ assay. **b** Viral titre of each homogenised lung lobe aliquot wase enumerated using a $TCID_{50}$ assay, the fraction representing the number of major lung lobes with detectable viral titres (above LOD) per ferret. Each dot represents the titre in a single lung lobe and the line shows the mean. Error bars represent one SD. LOD is shown as a dotted horizontal line, all values beyond LOD equal zero viral titre. Adjusted $p$-values and asterisks show statistically significant ($p < 0.05$) differences (Two-way ANOVA with Sidak's multiple comparisons).

of each virus when assessed at a high or low MOI, although the PA-I38T single mutant appeared to have a slightly delayed replication in the single-stage experiment (Fig. 1). Pyrosequencing analysis also confirmed that each virus maintained (at 95-100%) their relevant substitution at AA positions, NA-275 and PA-38, at 102 h post infection, indicating none of the mutants had reverted to the WT genotype (Supplementary Table 3). Overall, double and single mutant virus replication profiles remained similar to WT.

**Ferret in-host replication fitness of the double mutant clinical isolate virus.** To determine the replication fitness of the double mutant virus (NA-H275Y + PA-I38T) in the upper and lower respiratory tract (U/LRT), ferrets were infected with either WT or the double mutant NA-H275Y + PA-I38T isolates. Nasal washes were collected daily for 9 days post-inoculation to quantify the viral load in the URT. Based on infectious virus load in the URT, the replication of both viruses was comparable for the first 3 days following infection, but on days four, five and six, the mean viral load in the URT of animals infected with double mutant NA-H275Y + PA-I38T virus was significantly lower (by 1.59, 1.75, and 2.73 $\log_{10}TCID_{50}/$mL on day 4, 5 and 6, respectively) when compared to ferrets infected with the WT virus. The duration of viral

shedding was 1 day shorter for NA-H275Y + PA-I38T infected animals (5 days, 4/4 ferrets) than WT infected animals (6 days, 4/4 ferrets). Area under the curve (AUC) analysis of mean nasal wash viral titres revealed NA-H275Y + PA-I38T infected ferrets had a lower AUC ($17.98 \pm 1.43$) than WT infected ferrets ($25.77 \pm 1.29$) (Fig. 2a). While infectious viral titres shed from the URT were significantly different between the two isolates, quantitative RT-PCR (qPCR) analysis was not (Supplementary Fig 2a).

To quantify the pulmonary viral load, the infectious viral titre was determined in the five major lung lobes at day five post-inoculation. Three of four ferrets in both infection groups (WT or NA-H275Y + PA-I38T) had infectious virus present in two or more lung lobes (Fig. 2b, Supplementary Fig 2c). qPCR analysis of each lung lobe was consistent with virus titration data, showing similar viral load between WT or NA-H275Y + PA-I38T inoculated ferrets (Supplementary Fig 2b). We also observed by WGS that the NA-H275Y and PA-I38T mutations were unchanged in the virus samples recovered from ferret lungs, and no other amino acid changes from virus inoculum were detected (sequences as per Supplementary Table 1). Overall, virus titres detected in the lungs of WT and NA-H275Y + PA-I38T infected ferrets were comparable. We also compared infected ferrets for clinical signs, including weight and body temperature, which was monitored once daily, which revealed no significant

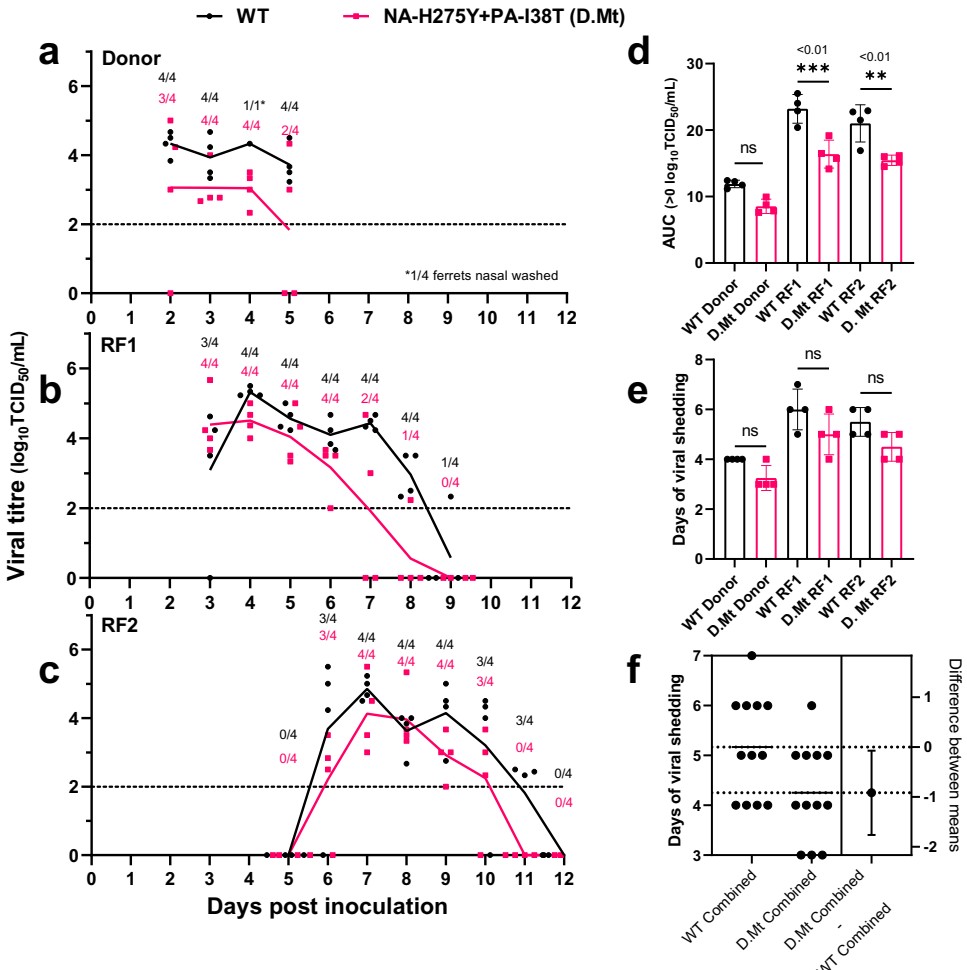

**Fig. 3 Infectious viral shedding in nasal washes from ferrets infected with WT or NA-H275Y + PA-I38T clinical isolates and in ferrets infected by airborne virus exposure.** Donor ferrets ($n = 4$) were infected by the intranasal route with 5 $\log_{10}TCID_{50}$/mL of pure WT or NA-H275Y + PA-I38T clinically isolated virus and co-housed with naïve recipient ferret 1 (RF1) separated by an airborne virus-permeable barrier at day 1 (D1) post inoculation. Infected RF1s were then co-housed with RF2 ferrets, enabling a second airborne transmission event to occur. Viral titre mean from nasal washes from donor (**a**), RF1 (**b**) and RF2 ferrets (**c**), by $TCID_{50}$ assay. Fractions above each symbol represent the number of ferrets shedding virus above LOD (dotted horizontal line), all values below LOD equal zero. No statistically significant (adjusted $p < 0.05$) differences was observed (Sidak's Two-way ANOVA multiple comparisons). **d** Area under the curve analysis was performed for each individual ferret ($n = 4$) and overall viral shedding was compared between WT (black) and NA-H275Y + PA-I38T (D.Mt, red). **e** Duration of infectious viral shedding above LOD was tallied for each ferret. **f** Estimation plot of the combined duration of viral shedding for all donor, RF1 and RF2 ferrets was analysed by Unpaired $t$-test to display a 95% confidence interval of the difference between the mean duration of shedding for WT and NA-H275Y + PA-I38T infected ferrets. Error bars indicate one SD. Statistically significant differences ($p < 0.05$) were compared by Tukey's Ordinary One-way ANOVA.

differences between WT and NA-H275Y + PA-I38T infected animals (Supplementary Fig 3).

**Ferret airborne transmission study**. Airborne transmission chains were initiated by inoculating donor animals that were housed adjacent to naïve recipient ferrets (RF) but separated by a perforated metal barrier, allowing the transfer of airborne virus only. For WT and NA-H275Y + PA-I38T viruses, transmission events from donor to the first recipient ferret (RF1) and RF1 to the second recipient ferret (RF2) were detected by nasal wash qPCR analysis cycle threshold <30 (*matrix* gene Ct value) within 2–3 days post exposure. The mean infectious virus titres shed for each group of ferrets is shown in Fig. 3a–c (individual ferret nasal wash titres shown in Supplementary Fig 4). The AUC analysis showed that overall infectious virus titres were lower in the NA-H275Y + PA-I38T infected ferrets compared to WT, with an AUC reduction of 28% in donors ($p = 0.13$), 29% in RF1 ($p = 0.0005$) and 26% in RF2 ($p = 0.0038$) (Fig. 3d), reinforcing the data from the previous

experiment (Fig. 2a). Although statistically significant differences were not observed for the duration of viral shedding between WT and the NA-H275Y + PA-I38T infected animals (Fig. 3e), a consistent trend of ~1 day shorter duration of viral shedding was observed in ferrets infected with NA-H275Y + PA-I38T compared to WT infected ferrets, across donor, RF1 and RF2 animals (Fig. 3f). The qPCR analysis revealed closely matched viral RNA load in both WT and NA-H275Y + PA-I38T infected ferrets over time (Supplementary Fig 5).

The WT and NA-H275Y + PA-I38T infected ferrets were also assessed for clinical signs such as weight loss and body temperature. RF1 and RF2 animals showed no significant difference in weight loss or body temperature during WT or NA-H275Y + PA-I38T influenza virus infection. One of the four NA-H275Y + PA-I38T infected animals suffered more severe weight loss and was euthanised due to its condition on D11 (Supplementary Fig 6).

Ferret nasal washes with detectable virus (>2 $\log_{10}TCID_{50}$/mL) from the transmission study were analysed by pyrosequencing to

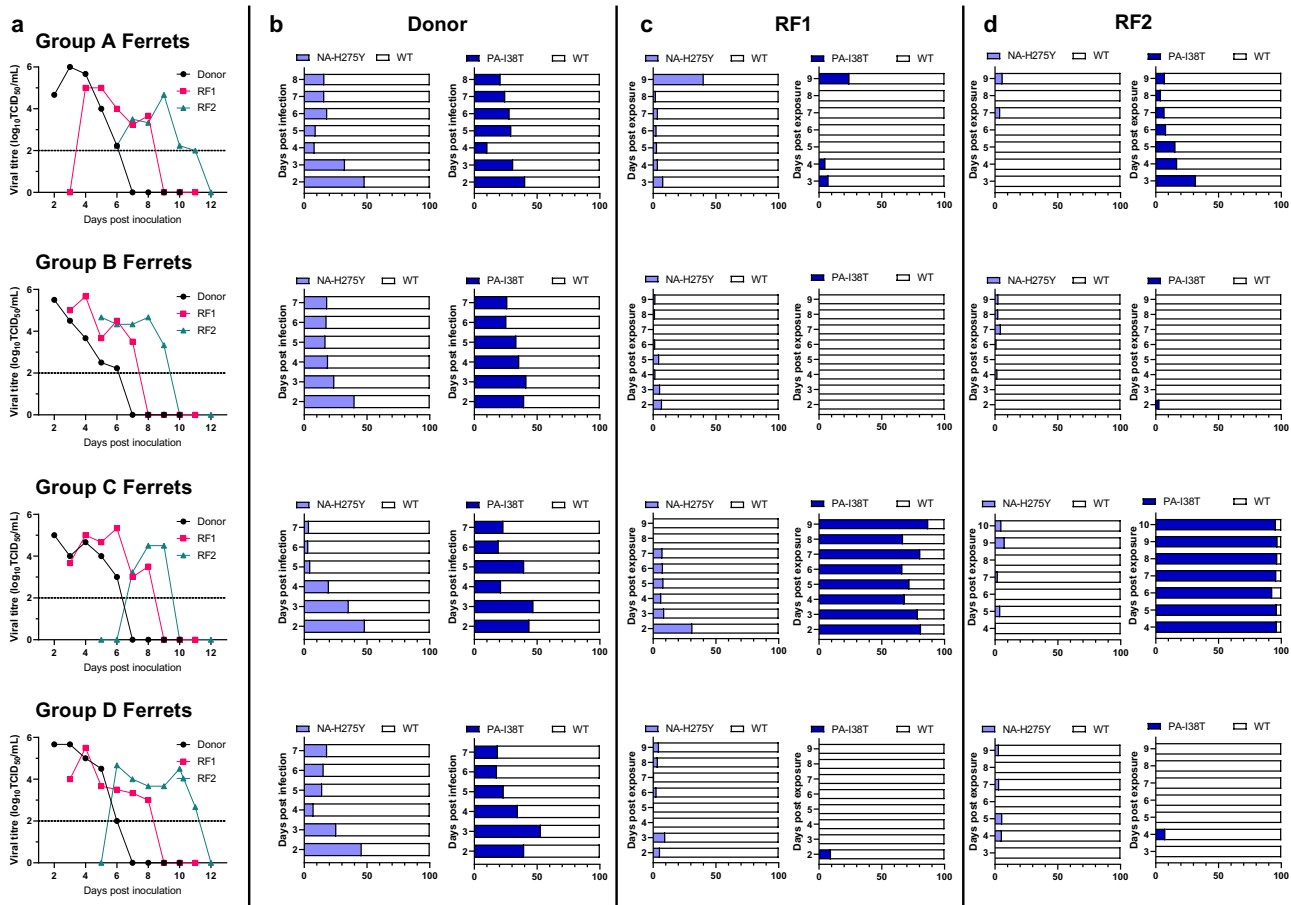

**Fig. 4 Infectious virus titres and pyrosequencing analysis of ferret nasal washes from 50:50 WT:NA-H275Y + PA-I38T clinical isolate inoculated donor and airborne transmission infected recipient ferrets.** Donor ferrets ($n = 4$) were infected via the intranasal route with 5 $\log_{10}\text{TCID}_{50}$/mL with 50:50 WT: NA-H275Y + PA-I38T clinically isolated virus and co-housed with naïve recipient ferrets (RF1) virus separated by an airborne virus permeable barrier, infected RF1s were then co-housed with RF2 ferrets. Nasal wash samples were titrated by $\text{TCID}_{50}$ assay for each Group A-D Donor-RF1-RF2 transmission pairs, LOD is shown as a dotted horizontal line, and all values beyond LOD equal zero viral titre (**a**). The relative proportion of virus encoding WT PA (white bars) and PA-I38T (dark blue bars) and WT NA (white bars) and NA-H275Y (light blue bars) were determined by pyrosequencing analysis and are shown side-by-side in bar charts for each donor (**b**), RF1 (**c**) and RF2 (**d**) ferret.

determine the viral genotype at the PA-38 and NA-275 AA positions. Sequence data from the nasal washes infected with WT virus confirmed that all four WT infected donor ferrets transmitted approximately 100% PA-I38 + NA-H275 virus through to RF1 and onward to RF2 animals (Supplementary Fig 7a). NA-H275Y + PA-I38T infected ferrets also shed virus with sequences consistent with the inoculum genotype, at approximately 100% PA-I38T + NA-H275Y, with the exception of Group I RF1 ferrets whose nasal washes had a mixed proportion of WT and double mutant virus genotypes, which resulted in the infection and shedding of pure (approximately 100%) WT virus to the next recipient ferret (Group I RF2) (Supplementary Fig 7b).

**Competitive mixture airborne transmission study.** To explore more subtle differences between the replicative and transmission fitness dynamics of the NA-H275Y + PA-I38T and WT viruses, a competitive mixture airborne transmission ferret study was performed with an initial challenge inoculum mixture of both WT and NA-H275Y + PA-I38T at an approximate 50%:50% ratio (based on pyrosequencing analysis). The viral titres from daily nasal washes demonstrated successful donor infection and onward transmission for each group of ferrets (Fig. 4a,

Supplementary Fig 8). To measure the head-to-head fitness, nasal washes with detectable virus by $\text{TCID}_{50}$ were analysed by pyrosequencing at the PA-38 and NA-275 loci to determine the proportion of WT or mutant virus. Pyrosequencing analysis of the donor ferret inoculum showed that instead of the desired 50:50 mixture, a mixture of approximately 60:40 WT:NA-H275Y + PA-I38T viruses was given to the donor ferrets. All four donor nasal washes showed a similar proportion of WT:NA-H275Y + PA-I38T to that of the inoculum at day 2 post-inoculation (bottom bar of each donor bar chart of Fig. 4b), however the proportion of WT gradually increased to become the dominant virus shed from donor ferrets (approximately 90% WT) by the last day of viral shedding (day 7 or 8). Donor to RF1 airborne transmission seeded approximately 100% pure WT virus in three of the four recipient ferrets, which resulted in three pure WT infected RF2 animals. The remaining Group C RF1 animal shed a mixed virus population at day 2 post exposure, which resulted in the shedding of pure (90-100%) single mutant NA-H275 + PA-I38T virus from both Group C RF1 and RF2. Overall, the pyrosequencing data revealed a competitive fitness advantage of WT over H275Y + PA-I38T virus within the donor ferrets, and after onward transmission the proportion of WT virus increased to 100% in 3 out of 4 of the RF1 and RF2 ferret transmission groups (Fig. 4c–d).

**Table 1 Pyrosequencing analysis at NA-275 and PA-38 of individual plaque picks grown from clinical isolate, ferret inoculum and representative nasal wash samples from the competitive mixture airborne transmission study.**

| Virus sample | Number of plaques picked (and percentage of total plaques) with one of four possible genotypes as determined by pyrosequencing | | | |
| --- | --- | --- | --- | --- |
| | WT | NA-H275Y + PA-I38T | PA-I38T | NA-H275Y |
| **Purified clinical isolate** | | | | |
| **Pre-treatment WT** | 92/92 (100%) | 0/92 (0%) | 0/92 (0%) | 0/92 (0%) |
| **Post-treatment NA-H275Y + PA-I38T** | 0/93 (0%) | 93/93 (100%) | 0/93 (0%) | 0/93 (0%) |
| **Competitive mixture airborne transmission study** | | | | |
| **50:50 WT: Double mutant mixture inoculum** | 57/74 (77%) | 17/74 (23%) | 0/74 (0%) | 0/74 (0%) |
| **Group C** Donor Day 2 | 27/65 (42%) | 15/65 (23%) | 17/65 (26%) | 6/65 (9%) |
| RF1 Day 2 | 8/59 (14%) | 19/59 (32%) | 32/59 (54%) | 0/59 (0%) |
| RF2 Day 5 | 16/48 (33%) | 0/48 (0%) | 32/48 (67%) | 0/48 (0%) |
| **Group B** Donor Day 2 | 26/42 (62%) | 4/42 (10%) | 7/42 (17%) | 5/42 (12%) |
| RF1 Day 2 | 48/48 (100%) | 0/48 (0%) | 0/48 (0%) | 0/48 (0%) |

**Genotype analysis of ferret nasal washes**. Pyrosequencing analysis unexpectedly revealed the rapid emergence of a PA-I38T single mutant virus from the 50:50 WT:NA-H275Y + PA-I38T inoculated Group C donor to the RF1 animal (Fig. 4). To investigate the origin of the single mutant PA-I38T virus further, individual virus plaques were grown, picked and analysed by pyrosequencing at positions NA-275 and PA-38 to determine the proportion of the WT, single mutant or double mutant viruses present in nasal washes from the ferrets. The genotype of 42 to 92 distinct plaques from representative Group C nasal wash samples were analysed, as well as two distinct Group B nasal wash samples, as a comparison group that did not demonstrate single mutant virus genotype emergence. Firstly, the WT and NA-H275 + PA-I38T purified virus stocks and 50:50 mixed inoculum were confirmed to be pure virus populations that did not contain detectable PA-I38T single mutant virus. However, analysis of two 50:50 WT:NA-H275 + PA-I38T inoculated donor ferret nasal washes (at day 2 post inoculation) revealed that Group C had a mixture of 17/65 (26%) plaques with a PA-I38T single mutant genotype and 6/65 (9%) NA-H275Y single mutants and Group B also had a mixed virus population with all four possible genotypes detected. Therefore the origin of PA-I38T single mutant infection in the Group C ferrets was likely to have been from the donor ferret, which resulted in 32/48 (67%) PA-I38T genotype plaques by day 5 in RF2 (Table 1).

The above results suggest that reassortment events between the WT and double mutant virus may be the source of the single mutant NA-H275Y and PA-I38T viruses generated from 50:50 mixture inoculated donor ferrets. To investigate if reassortment occurred, influenza virus whole-genome sequencing (WGS) was performed on the ferret inoculum samples and representative nasal washes, capturing the potential genetic changes occurring from donor to RF1 to RF2 within at least one transmission group of each pure and mixed inoculum experimental group. All sequenced viruses were genetically identical apart from five AA changes on four genes (including NA-H275Y, PA-I38T, PA-P325Q, HA-A204T and PB1-Y129H), which differed between the WT and NA-H275Y + PA-I38T plaque purified clinical isolates. The five AAs of interest remained stable throughout transmission events of the pure WT Group G and the pure NA-H275Y + PA-I38T Group L infected ferrets. However, the pure NA-H275Y + PA-I38T Group I infected ferrets (where the RF1 viral infection became a mixture with WT virus, potentially as a result of contamination) revealed evidence of reassortment with the WT virus, given the RF2 shed pure virus with a WT sequence on the PA and NA genes (PA-I38, PA-P325, NA-H275), and double mutant virus associated sequence on the HA and PB1 genes (HA-A204 and PB1-Y129). Reassortment also occurred in the 50:50 WT:NA-H275Y + PA-I38T inoculated Group B and C ferrets, confirming the reassortment of gene segments from both virus inoculums in the Group C transmission group that resulted in a dominant single mutant PA-I38T virus infection (Fig. 5, Supplementary Table 4).

## Discussion

In this study we evaluated the replicative fitness and transmissibility of influenza A(H1N1)pdm09 virus isolates derived from an immunocompromised patient that developed dual resistance to both PER/OST and BXM after initially being treated with PER monotherapy for 1 day before starting PER and BXM combination therapy[31]. Three respiratory samples were obtained from this patient and four viruses were isolated from these samples; a WT isolate with no antiviral resistance mutations (NA-H275 and PA-I38), separate single resistance mutation isolates with either NA-H275Y (causing PER/OST resistance) or PA-I38T (causing

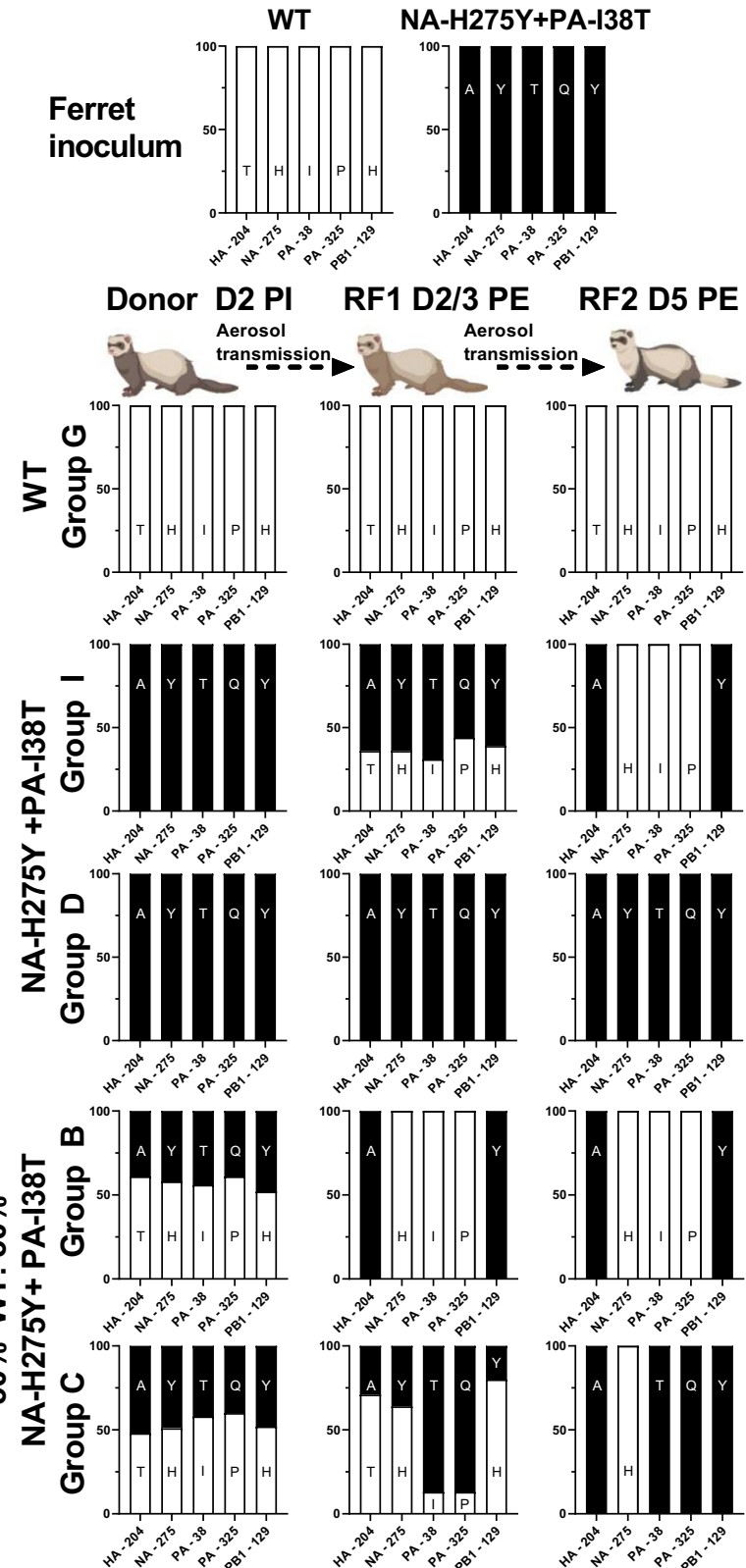

**Fig. 5 Summary of whole-genome sequencing on ferret inoculum and nasal wash samples from pure WT, pure NA-H275Y + PA-I38T and 50% WT: 50%NA-H275Y + PA-I38T isolate infected ferrets.** Ferret inoculum and nasal wash samples post inoculation/exposure (PI/E) from pure WT, pure NA-H275Y + PA-I38T or 50% WT: 50%NA-H275Y + PA-I38T virus infected ferrets were sequenced using WGS. Five amino acid (AA) changes (across the HA, NA, PA and PB1 genes) distinguished ferret inoculum WT and NA-H275Y + PA-I38T virus genotypes. The bar charts display the proportion of AA codon changes detected by NGS for each sample associated with the WT (white) and NA-H275Y + PA-I38T (black) virus inoculum for each gene loci (along X-axis; HA-294, NA-275, PA-38, PA-325, PB1-129). Percentage frequency of AA codons is determined from variant bases called with a read depth over 1000 and frequency of greater than 5%. Created with BioRender.com.

BXM resistance), and an isolate with both resistance mutations NA-H275Y + PA-I38T (causing resistance to both PER/OST and BXM). Testing of these isolates indicated that while the dual-resistant isolate had comparable replicative fitness to the WT isolate in vitro, it replicated to lower viral titres and was cleared more rapidly from the upper respiratory tract of ferrets compared to the WT isolate. However, when viral titres in the lungs were examined on day five post-inoculation, no clear difference was observed between the NA-H275Y + PA-I38T and WT infected ferrets, with similar levels of pulmonary viral load as previously reported with influenza A(H1N1)pdm09 viruses in ferrets[32–34].

Airborne transmission is the most common form of influenza virus spread between humans in the community[35,36]. We utilised the well-established ferret airborne transmission model, which relies on respiratory droplet transfer of viruses between animals, and requires a higher viral transmissibility to be achieved compared to direct contact transmission models[37]. Previous ferret studies have characterised A(H1N1)pdm09 viruses with either OST-resistant NA-H275Y or BXM-resistant PA-I38T mutations independently. Viruses with NA-H275 demonstrated a robust within-host replicative fitness and transmission potential[38,39], but some studies showed a reduced airborne transmission capacity[40,41]. When assessed as pure virus population, viruses with PA-I38T showed only a slight reduction in upper respiratory tract viral shedding and had equivalent airborne transmission capacity compared to WT viruses[42–44]. The double mutant NA-H275Y + PA-I38T virus isolate in our study showed successful airborne transmission to all recipient ferrets, but the nasal washes had reduced viral titres (as measured by area under the curve analysis of TCID$_{50}$ results) and the virus was cleared a day earlier compared to the WT virus isolate. In contrast, the viral RNA quantification indicated no substantial difference in viral loads detected in the URT between these groups, however, this metric may be less indicative of viral fitness than TCID$_{50}$ assay data, given it also measures non-infectious virions.

Genetic analysis confirmed that the NA-H275Y + PA-I38T virus remained stable and did not revert to WT during in vitro and in vivo ferret studies, except for one group of ferrets infected with the double mutant virus, where the WT virus became dominant in RF1 and RF2. This outcome may have resulted from low-level contamination with the WT virus, or perhaps reversion at both loci, although this would seem an unlikely event. Previous studies have also shown that both A(H1N1)pdm09 and A(H3N2) viruses with PA-I38T had good stability in ferret transmission studies[43,44], but have shown limited person-to-person transmission[23,45]. Additionally, NA-H275Y stability has been demonstrated by widespread community transmission of the previous seasonal A(H1N1) virus in 2008-2009[8,9] and localised transmission in A(H1N1)pdm09 viruses[46–48].

The competitive mixture ferret airborne transmission study demonstrated a consistent within-host fitness reduction for the NA-H275Y + PA-I38T virus. During these transmission events the co-administered NA-H275Y + PA-I38T virus was clearly outcompeted following the initial donor to recipient ferrets virus transmission, indicating that WT virus was transmitted preferentially during the viral airborne transmission bottleneck from Donor to RF1, as has been described by Varble et al.[49]. Similar competitive mixture experiments with single mutant PA-I38T and WT A(H1N1)pdm09 isolates have previously shown no significant within-host fitness cost, but a drastic between-host fitness cost compared to WT[50,51], while single mutant NA-H275Y and WT competitive mixture studies have shown no significant within-host fitness cost and variable between-host fitness cost with reports of no reduction[52,53] or some reduction[40].

Possible mechanisms that might explain the reduced competitive fitness of these mutant viruses include reduced enzymatic activity, changes in structural stability, or for the PA-I38T, increased genome replication errors resulting in additional defective interfering virus particles (DIVP). Enzymatic efficacy studies of the PA-I38T mutation have shown that this change reduced endonuclease enzymatic activity[15,43], although polymerase complex activity was unchanged (or higher), compared to the WT virus polymerase[43,51]. Computational structural analysis of NA-H275Y may indicate impaired stability compared to WT, although this virus also contains the NA mutations N369K and V241I, which were demonstrated by Hurt et al. to restore stability[46]. Our study did not find any differences in the proportion DIVP in the nasal wash samples from WT or NA-H275Y + PA-I38T infected ferrets as determined by WGS (looking for large deletions in individual gene segments). Hence, the true mechanism for the observed decrease in competitive fitness of this dual-mutant virus is likely to be multifaceted that may be difficult to fully elucidate.

Further NGS analysis of one of the outliers in the competitive mixtures study, where an RF1 ferret in Group C (Fig. 4) generated a predominantly single mutant PA-I38T infection (which was then transmitted effectively to RF2), showed that reassortment occurred between the WT and NA-H275Y + PA-I38T virus mixture, following the initial infection of the Group C donor ferret. This was apparent as the two substitutions detected in PA (I38T and P325Q) were always detected in identical proportions, indicating that a whole gene segment reassortment event had occurred and not a site-specific recombination or reversion event. Similar reassortment events have been described previously by Goldhill et al., who also observed the emergence of a single mutant viruses from a ferret inoculated with a mixture of WT and a favipiravir-resistant double mutant influenza A(H1N1)pdm09 virus[54]. Another study by Richard et al.[55] detailed rapid influenza virus reassortment across all eight genes within co-infected URT cells of ferrets with a mixture of WT and a mutant A(H1N1)pdm09 virus with a silent substitution tagging each influenza virus gene. Their study also confirmed that reassortment only occurred when WT and variant-labelled influenza viruses were inoculated into the same anatomical compartment of the ferret respiratory tract[55]. These studies, alongside our results, suggest that reassortment is likely to have occurred in all four WT and double mutant virus co-infected donor ferrets, but despite these events, the WT virus emerged as the dominant virus in the presence of both double and single mutant (NA-H275Y or PA-I38T) viruses in 3 out of 4 ferrets.

The potential for the future emergence and widespread circulation of this dual PER/OST and BXM-resistant virus is unknown, but given that NA-H275Y + PA-I38T variant viruses have not been observed outside of the two immunocompromised patients in the FLAGSTONE study[31] it would seem unlikely, although as yet, little PER/OST and BXM combination therapy has been used. While the double mutant virus may be capable of person-to-person transmission, based on successful airborne transmission between ferrets in this study, the reduced relative fitness observed in ferrets compared to a WT virus, suggests that this double mutant virus would be unlikely to replace an existing A(H1N1)pdm09 WT virus that was circulating in the human population.

Limitations of this study include the heterogeneity present amongst the outbred ferrets used in these experiments and the small number of animals infected with each virus per group, making it difficult to elucidate subtle differences in viral kinetics. Additionally the peak viral loads in the ferret lungs may not have been fully captured at day five post inoculation, and further time points might be more informative. Infections of the lungs is important as it occurs in 30-40% of hospitalised patients with influenza A(H1N1)pdm09 infection, and is often associated with more severe disease[56]. It is also worth noting that while ferrets are the best available animal model for studies of influenza transmission, no animal model

perfectly recapitulates influenza disease in humans[57]. Our study also did not measure the fitness of these double mutant isolate viruses in the presence of antiviral drugs, but it would be assumed that their relative fitness would be enhanced compared to drug-sensitive virus in the presence of the relevant antiviral(s).

In the future, patients receiving combination therapy with OST/PER and BXM or other influenza antiviral combinations, should be closely monitored for the emergence of any drug resistant viruses. Early studies suggest that combination therapy of a neuraminidase inhibitor and BXM may reduce the incidence of resistance mutations compared to monotherapy[31]. This is also supported by human clinical data using OST and the unlicensed polymerase inhibitor pimodivir in combination therapy[58]. Therefore, combination therapy may be beneficial in hospitalised and in immuno-compromised patients, where influenza virus is shed for longer periods and resistance mutations are more likely to emerge[59–61]. Dosing of both antivirals simultaneously from the start of treatment (unlike that given to the patient in this report) should reduce the risk of generating viruses with dual-resistance, and may also shorten the patient's duration of virus shedding and symptoms compared to single or sequential antiviral therapies. This study highlights the need for continued influenza virus surveillance for drug-resistant viruses so that public health authorities can readily adapt influenza antiviral recommendations to ensure that the circulating influenza viruses remain susceptible to the antiviral drugs in use. The emergence of dual-drug resistant influenza viruses shows that continued development of new antiviral drugs against influenza might still be required for the future.

## Materials and methods

**Cells**. Madin-Darby Canine Kidney (MDCK CCL-34) cells (ATCC, USA) were cultured at 37 °C and 5% $CO_2$ in Dulbecco's Modified Eagle Medium (DMEM, high glucose pyruvate; Gibco, USA). The DMEM was supplemented with 10% foetal bovine serum (Bovogen Biologicals, Australia), 1x GlutaMAX (Gibco, USA), 1x MEM non-essential amino acid solution (Gibco, USA), 0.05% sodium bicarbonate (Gibco, USA), 20 μM HEPES (Gibco) and 100 U/mL penicillin-streptomycin solution (Gibco, USA). DMEM media was used for virus dilution, containing the above constituents excluding only serum, hereafter known as maintenance media. Maintenance media was also used for virus infection protocols, although media was supplemented with 8 μg/mL TPCK-treated trypsin (SAFC Biosciences, USA), hereafter known as infection media.

**Viruses**. Influenza A(H1N1)pdm09 virus isolates were derived from clinical samples collected from one subject at days 1, 4 and 10 (d1, d4 and d10) post enrolment into a phase III clinical study (clinicaltrials.gov identifier NCT03684044[31]). To ensure pure virus populations, the three virus samples were subjected to two rounds of plaque purification and following expansion in MDCK cells were sequenced to confirm the required genotypes had been obtained without mixed bases. These purified isolates were given the following designations; A/South Korea/90207_d1_ A/2020 (Day 1; WT); A/South Korea/90207_d1_B/2020 (Day 1; NA-H275Y); A/South Korea/90207_d4_C/2020 (Day 4; PA-I38T); A/South Korea/90207_d10_D/2020 (Day 10; NA-H275Y + PA-I38T) (Supplementary Table 1). All virus aliquots were stored at −80 °C until used. The infectious virus titre was determined prior to use (described below).

**Virus plaque purification assay**. MDCK cells seeded at $1.05 \times 10^6$ cells/3 mL in 6-well plates overnight before plates were washed twice with phosphate buffered saline (PBS). Virus was diluted 10-fold from $10^{-2}$ to $10^{-7}$ and 0.5 mL of inoculum was added to each well before incubation for 1 h at 35 °C/5% $CO_2$. The virus inoculum was removed and 2 mL overlays of infection media containing 0.5% agarose were added. Plates were incubated for 3 days before distinct plaques were isolated with a pipette tip and diluted in 500 μL of maintenance media for virus propagation or genotypic analysis.

**Virus titration assay**. Infectious virus titres were determined by 50% tissue culture infective dose ($TCID_{50}$) assay in MDCK cells, as previously described[62]. In brief, MDCK cells were seeded at $3 \times 10^4$ cells/100 μL into a 96-well plate and cultured overnight at 37 °C/5% $CO_2$. 10-fold serial dilutions of virus samples were overlayed on confluent MDCKs cells following two washes with PBS, on a flat-bottom 96-well plate. Plates were incubated for 1 h at 35 °C/5% $CO_2$ and then the virus inoculum was removed from the plates and replaced with 200 μL infection media. The plates were incubated for 3 days and virus growth was determined by red blood cell agglutination (1% turkey red blood cells in PBS). The viral titre was determined by the Reed-Muench method[63].

**Whole-genome sequencing**. Viral RNA was extracted from a 140 μL virus sample using the QIAamp viral RNA mini kit (Qiagen, Australia). The eight influenza gene segments were then amplified by one-step RT-PCR with universal influenza A primers[64] using the qSCRIPT XLT kit (Quanta). Viral DNA libraries were assembled and barcoded using the Nextera library preparation kit according to the manufacturer's instructions. Sequencing was performed on the Illumina iSeq 100 platform. Downstream analysis of sequencing data was performed by in-house WHOCCRRI, Melbourne pipeline, outlined previously by Lee, et al.[44], based on the CDC IRMA pipeline[65]. Consensus genotype analysis performed using Geneious Prime (Biomatters, Auckland). Variant call format was used to quantify mixed base genotypes, with a minimum variant calling threshold of 1%.

**Antiviral drugs**. A fluorometric neuraminidase inhibition assay was performed to analyse virus susceptibility against four NAIs, oseltamivir carboxylate, peramivir (BCX-1812), laninamivir (R-125489) and zanamivir (kindly provided by Hoffman-La Roche Ltd, Switzerland; BioCryst Pharmaceuticals, USA; Daiichi-Sankyo, Japan and GlaxoSmithKline, Australia, respectively). 300 μM of NAI stock solution was prepared in 2x Assay Buffer (AB) (66.6 mM MES and 8 mM $CaCl_2$ pH 6.5) and stored at −20 °C for up to 12 months.

A cell-based baloxavir susceptibility assay was performed with Baloxavir acid (S-033447), an active form of BXM, which was kindly provided by Shionogi & Co. Ltd., Japan. 10 mM BXA stock solution was prepared in DMSO and stored at −80 °C.

**Neuraminidase inhibition assay**. Briefly, each clinical isolate virus was titrated to its optimum NA activity as described by Leang and Hurt[66]. 50 μL of diluted virus was incubated with 50 μL of oseltamivir carboxylate, peramivir, laninamivir or zanamivir at a range of concentrations (0.03–30,000 nM) for 45 min at room temperature. 50 μL of 300 μM MUNANA (2'-(4-Methylumbelliferyl)-α-D-N-acetylneuraminic acid) substrate was added to the virus and drug mixture and further incubated for 60 min at 37 °C. To terminate the reaction, 100 μL of 0.14 M NaOH in ethanol was added. The NA enzymatic activity at each NAI concentration was read using a fluorometer. JASPR v1.2 curve fitting software (kindly provided by Dr. Larisa Gubareva, CDC, USA) was used to determine the half-maximal inhibitory concentration ($IC_{50}$) of each virus.

**Cell-based BXA susceptibility assay**. The susceptibility of clinical influenza viruses to BXA was determined using a cell-based assay described by Koszalka et al.[67], with modifications. Briefly, all viruses were initially titrated to a standard optical density (O.D.) of 1.5–2.0. Each virus was titrated by half-log dilutions in maintenance media. MDCK cell monolayer, prepared at concentration $2.5 \times 10^4$ cells/well in a 96-well plate on a day prior, was washed twice with PBS and 50 μL of diluted virus was added to the cells and incubated for 90 min at 35 °C, 5% $CO_2$. 50 μL of 2x Infection Media (DMEM maintenance media with 2% Bovine Serum Albumin (BSA) (Sigma-Aldrich), 4 μg/mL TPCK-trypsin (SAFC Biosciences)) was added to the plate and incubated for 35 °C, 5% $CO_2$ for 24 h. Virus was removed and the cell monolayer was washed three times with PBS and fixed with 100 μL of chilled 80% acetone for 15 min. Plates were dried at room temperature before washing with Wash Buffer (0.3% Tween20 (Sigma-Aldrich) in PBS) three times. 100 μL of influenza A nucleoprotein antibody MAB8251 (Millipore, USA) in 5% skim milk was added to the plate and was left overnight at 4 °C. On the following day, the monolayer was washed three times with Wash Buffer and incubated with 100 μL goat anti-mouse IgG-horse radish peroxidase (Biorad, US) for 1 h at room temperature. The plate was washed five times with Wash Buffer before adding 100 μL of TMB microwell peroxidase substrate system (SeroCare) to each well. The plate was incubated in the dark for 10 min at room temperature and 100 μL of 1 M Hydrochloric acid was added. The O.D. of each well was read at 450 nm using a colorimeter. To determine the virus dilution factor, a graph of O.D. against half-log virus dilutions was plot using GraphPad Prism 9.

To determine the susceptibility of the virus to BXA, each virus was diluted to 1.5–2.0 O.D. in DMEM maintenance media and 50 μL was added to the MDCK cell monolayer (prepared a day prior as above). After 90 min of incubation at 35 °C, 5% $CO_2$, 50 μL of BXA in 2x Infection Media at a range of concentrations 0.01–12,800 nM was added to the well. After 24 h of incubation, the monolayer was washed, fixed, stained and the O.D. of each well was read, as before. Based on the O.D. reading, the percentage of inhibition was calculated. The half-maximal effective concentration ($EC_{50}$) of BXA was determined using GraphPad Prism 9.

**Single-stage virus replication assay**. MDCK cells were seeded overnight at $6 \times 10^4$ cells/2 mL in 24-well plates. Confluent MDCKs were infected at a MOI of 5 for 1 h at 35 °C with virus isolates. The inoculum was then removed and replaced with 2 mL of infection media. Virus was harvested from each well at 2-, 4-, 6-, 8-, 10- and 12-h post infection. Samples were stored at −80 °C before virus titre quantification.

**Multi-stage virus replication assay**. Confluent T-25cm² flasks of MDCK cells were inoculated at an MOI of 0.01, and incubated for 1 h at 35 °C. The inoculum

was then removed and replaced with 10 mL of infection media. Virus was harvested from each flask at 6-, 24-, 30-, 48-, 54-, 72-, 78-, 96- and 102-h post infection. Supernatant aliquots were stored at −80 °C before virus titre quantification.

**Ferrets**. Outbred male and female ferrets (*Mustela putorius furo*) were obtained from commercial breeders (Animalactic Animals & Animal Products Pty Ltd, Australia) and were a minimum of 12 weeks of age and 0.6 kg in body weight. Seronegativity against the four different types/subtypes of recently circulating human influenza virus strains was confirmed by haemagglutination inhibition assay. Ferrets were housed individually in high efficiency particulate air filtered cages with ab libitum food, water and enrichment equipment throughout the experimental period. All animal procedures conducted in this study were approved by the University of Melbourne Animal Ethics Committee (project license no. 20033) in accordance with the Australian Government, National Health and Medical Research Council Australian code of practice for the care and use of animals for scientific purposes (8th edition).

**Ferret challenge study**. The ferret in-host replication fitness study included two groups of eight ferrets inoculated with either pure WT or NA-H275Y and PA-I38T patient-derived purified influenza A(H1N1)pdm09 isolates (*n* = 16). All ferrets received anaesthesia (1:1 (v/v) ketamine (100 mg/mL) and xylazine (20 mg/mL)) via intramuscular injection (IM) and 5 $\log_{10}TCID_{50}$/mL units of virus in 500 µL PBS was delivered by the intranasal route (250 µL per nostril).

Ferrets were sedated (Xylazine; 5 mg/kg), and subsequently monitored (clinical signs, subcutaneous microchip temperature and body weight) and nasal wash samples were collected with 1 mL of PBS daily, using methodology previously defined[44]. At day five post-inoculation, half of each group (four animals) were euthanised (Lethabarb; 0.5 mL/kg), and the five major lung lobes were excised, weighed, and each whole lobe was separately homogenised in 5 mL of PBS using a gentleMACS Octo Dissociator (Miltenyi Biotec). Residual cells and connective tissue were removed by centrifugation at 1600 rpm. The remaining eight ferrets were sacrificed 14 days post-inoculation, cardiac bleeds were obtained for analysis by haemagglutination inhibition to the infecting virus. The infectious viral load of ferret lung lobes and nasal wash samples, which were determined by $TCID_{50}$ titration assay, performed as outlined above.

**Real time RT-PCR**. Viral RNA quantitation was performed each day of ferret nasal wash (prior to storage at −80 °C), and ferret lung lobe aliquots were also analysed by the SensiFast Probe Lo-ROX One-Step qRT-PCR System Kit (Bioline), as outlined in previous study[44]. Influenza A cycle threshold (Ct) value was analysed by the ABI 7500 Real Time PCR System (Thermo, Australia) under the following conditions; 45 °C for 10 min, one cycle; 95 °C for 2 min, one cycle; 95 °C for 5 s then 60 °C for 30 s, 45 cycles. The influenza A RNA primers detected the *matrix* gene using a forward (5'-GACCRATCCTGTCACCTCTGAC-3'), reverse (5'-GGGCATTYTGGACAAAKCGTCTACG-3'), and Taqman probe (5'-6FAM-TGCAGTCCTCGCTCACTGGGCACG-BHQ1-3') sourced from CDC Influenza Division (Atlanta, United States of America).

**Ferret airborne transmission model experiments**. The experimental procedure for the transmission study is shown in Supplementary Fig. 9, with three different inoculums; 100% WT, 100% double mutant and a 50:50 mixture of WT:double mutant (Supplementary Table 4). In brief, donor ferrets were infected via intranasal inoculation with influenza virus diluted to 5 $\log_{10}TCID_{50}$/mL in 500 uL PBS (250 µL per nostril). After infection, all ferrets were monitored daily. The first group of naïve recipient ferrets (RF1s) were introduced into the donor-adjacent cage at 1 day post-inoculation, a double panel perforated metal sheet (two staggered panels are 25.4 mm apart with holes of 5 mm in diameter and spaced 3 mm apart) and one-directional circular airflow was used for airborne transmission studies, this cage design does not distinguish between airborne droplet and aerosol virus transmission[5]. At day 2 post-exposure to the donor animals, nasal washes from RF1 animals were collected and immediately analysed for viral RNA by qPCR. A Ct value of less than 30 indicated a ferret was influenza virus positive and RF1 was subsequently moved to a new cage adjacent to RF2, where both animals stayed for the remainder of the trial. All animals were sacrificed at day 14 post-PCR positive and cardiac bleeds were taken for serology analysis prior to euthanasia.

**Pyrosequencing**. Viral RNA was extracted, as above, from ferret nasal washes on the day of sample processing and stored at −80 °C. RT-PCR amplification of PA-38 and NA-275 gene regions was performed using biotin-tagged primers (Supplementary Table 5) and the MyTaq One-Step RT-PCR kit (Bioline, Australia). The pyrosequencing analysis was performed on amplified cDNA with sequencing primers (also described in Supplementary Table 5) and the PyroMarkQ96ID system (Qiagen). Sequence analysis revealed the relative proportion of wild type and SNPs at position PA-38 and NA-275, as previously outlined[68].

**Statistics and reproducibility**. All statistically analysis performed throughout was named with the statistical test used with information about the exact sample size (>3), any assumptions or corrections, and the resulting *p*-value of the null-

hypothesis tests. Standard deviation was used to capture error from the mean. In terms of general reproducibility, in vitro studies were conducted with three independent experiments performed in triplicate, and ferret studies were performed with four animals per group and reproduced once.

**Reporting summary**. Further information on research design is available in the Nature Research Reporting Summary linked to this article.

## Data availability

All data generated or analysed during this study are included in this published article (and its supplementary information files). Numerical source data for Figs. 1–5 has been made publically available in figshare with the identifier, https://doi.org/10.6084/m9.figshare.21008614[69]. The whole-genome sequence data of the clinically-derived viruses used in this study are available in GISAID with the identifiers A/South Korea/90207_d1/2020 (ID: 13655148), and A/South Korea/90207_d10/2020 (ID: 13655147).

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

## Acknowledgements

The authors thank Dr. Yi-Mo Deng, Jean Moselen, Xiaomin Dong and Dr. Ammar Aziz (WHO CCRRI, Melbourne) for whole-genome sequencing and analysis support of virus samples. Additionally, we would like to thank Heidi Peck, Monica Bobbit and Amalani Metuisela (WHO CCRRI, Melbourne) for titration assay support. We appreciate the support of the BioResources Facility at the Peter Doherty Institute for Infection and Immunity. The authors also acknowledge the patient enroled in NCT03684044 and their family, from whom these virus samples were derived from. The Melbourne WHO Collaborating Centre for Reference and Research on Influenza is supported by the Australian Government Department of Health. This work was funded by F. Hoffmann-La Roche (https://www.roche.com/).

## Author contributions

H.L.S. was co-lead investigator and prepared the original draft. E.J.M. was co-lead investigator. S.K.B. and P.K. performed integral investigations. S.W., T.S., S.K., S.O., K.B., K.K., and A.C.H. provided project resources and conceptualisation. I.G.B. supervised the project and conceptualisation. All authors contributed to review and editing.

## Competing interests

These authors declare the following competing interests: K.K, S.W., and A.C.H are employees of F. Hoffmann-La Roche. All other authors declare no competing interests.

## Additional information

 ns license, unless indicated otherwise in a credit line to the material. If material is not included in the article's Creative Commons license and your intended use is not permitted by statutory regulation or exceeds the permitted use, you will need to obtain permission directly from the copyright holder. To view a copy of this license, visit http://creativecommons.org/licenses/by/4.0/.

© The Author(s) 2022

