## [Peer Review File · Communications Biology]

Reviewers' comments:

Reviewer #1 (Remarks to the Author):

Stannard et al. reported characterization of a dual drug-resistant (NA-H275Y+PA-I38T) A(H1N1)pdm09 virus isolate and compared its replicative capacity, airborne transmissibility and relative fitness with those of the wild type and single mutant viruses (WT, NA-H275Y and PA-I38T), using both in vitro assays and a ferret model. It clearly demonstrated that this dual-mutant virus could efficiently transmit between ferrets via airborne exposure, but its viral fitness was moderately reduced compared with the WT virus. This is a carefully designed study with interesting, and quite informative results. I only have some minor comments:

- Lines 16-18: As the two patients who shed viruses with treatment-emergent dual-mutations (NA-H275Y+PA-I38T) were previously reported in reference [30], in the Abstract, it would be better to clarify "Previously, two patients... were found", and "Here/In this study,". This can help the readers to distinguish the results from these two studies.
- Line 95-96: Whether the PA-P325Q may potentially have a role in facilitating the emergence of PA-I38T or stabilizing the mutant was not known or tested. It would be prudent to delete the statement "and therefore is unlikely to play a role in antiviral resistance or the stability of the PA-I38T mutation".
- Line 112, " high and low" should be "high or low".
- Line 152, "...WT or the..." should be "...WT and the...".
- Lines 243-244: actually, the ferret model used here could not distinguish the droplet and aerosol transmissions. It would be better to use the term "airborne transmission".
- Line 344, reference [30] (rather than [29]) should be cited.
- Full form of "PBS" should be spelled out the first time you use it in Line 353, rather than in Line 362.
- Lines 356, 447 and 448: "uL" should be "µL".
- Lines 374 and 457, "performed" should be "was performed" or "were performed" respectively.
- Line 381: "and GlaxoSmithKline, Australia, respectively": "and" and ", respectively" should be added to this sentence.
- Line 384: " ..., which was kindly provided by ...": "which" was missed in the sentence.
- Lines 402-403, a ")" was missed after "(DMEM....TPCK-trypsin (SAFC Biosciences) ".
- Line 434, "Mustela putorius furo" should be italicized.
- Although many researchers use the term "post-infection", I think it would be more accurate to use "post-inoculation", as infections may not always immediately established after the "inoculation". Also, please be consistent in the use of this term throughout the main text, figures and supporting information.
- Line 694, "nasal" should be "nasally" (i.e. "... were nasally washed...").
- Line 695, "... ferrets that were culled....": "that" should be deleted.
- Lines 696-697, should be " (b) Viral titre of each ... was enumerated...".
- Line 700, "statistical" should be "statistically".
- Line 738, " were WGS" should be "were sequenced using WGS" or "were whole genomically sequenced".
- For a phrase with extremely long attributive nouns, such as "patient day 1 post enrolment virus consensus genome (line 25, Supporting information)", it may be more readable to transform it into something like "virus consensus genome of the patient's day 1 post enrolment sample". Please go through the manuscript and see if you can reduce the use of such kind of long attributive structures.
- Line 47 of Supporting information: "... virus isolates were nasal (should be "nasally") washed..."; "Half of the ferrets that were culled": "that" should be deleted.
- Line 91 of Supporting information: "multiple comparisons test illustrate..." should be "multiple comparison tests illustrate..."; Line 93 "depicts" should be "depict"; Line 94 "comparisons" should be "comparison".
- Line 119 of Supporting information: "... (was not) nasal washed..." should be "...nasally washed...".

Reviewer #2 (Remarks to the Author):

The authors have presented convincing data indicating that an A(H1N1)pdm09 variant with dual mutations (NA-H275Y and PA-I38T) resistant to both neuraminidase inhibitor and baloxavir marboxil had become less competent in infecting ferrets when co-administered with the WT virus, though the replicative fitness of the dual-mutant was similar to the WT virus in in vitro models. The findings are significant in terms of assessing the transmissibility of this dual mutant in community settings. The manuscript, however, focuses almost entirely on the description of the phenotypic observations of the double mutant without in-depth discussion on the possible mechanisms of the loss of viral fitness of the mutant in the ferret infection model with co-administered WT virus. Would it be the results of competition for the sialic acid receptors or efficiency in transcription of the viral genomes? Or would it be the differential amounts of defective viral particles generated by the mutant and the WT viruses? Or would it be the differential capability of the mutant and the WT viruses to escape host immune responses? Those mechanistic aspects are important to help the scientific community to gain further insights into the potential threat of these accumulating resistant mutations. The manuscript will be much improved if the authors may provide experimental results and speculations to explain the observed phenomenon.

Reviewer #3 (Remarks to the Author):

Stannard et al. "Assessing the fitness of a dual-antiviral drug resistant human influenza virus in the ferret model."

In this manuscript, the authors isolated influenza A/H1N1 2009 pandemic viruses with resistance to both neuraminidase inhibitors and baloxavir marboxil from immunocompromised patients and characterized the dual-resistant isolate in vitro and in vivo. Their results suggest that the dual-mutant influenza A/H1N1 2009 pandemic virus have a moderate loss of viral fitness compared to the wild-type virus.

The data presented in the manuscript are interesting and very important.

- Fig. 2b. Did the authors confirm that the NA-H275Y and PA-I38T mutations were retained in the viruses recovered from the lungs of NA-H275Y+PA-I38T-infected animals on Day 5 post-infection?
- Page 3 line 96. "hemagglutinin (HA)-A204T". The authors need to clarify whether this number means H1 or H3 numbering.
- Page 4 line 112. "MOI" needs to be spelled out on the first use.

15th July 2022

Dear Reviewers,

On behalf of my co-authors, I would like to submit the following revised manuscript entitled “Assessing the fitness of a dual-antiviral drug resistant human influenza virus in the ferret model” for publication in *Communications Biology*. We thank the reviewers for their detailed edits and comments. We found their insight thought provoking and beneficial to the quality of this publication.

Please see below the comments of each reviewer (in black) addressed by my co-authors (in red).

Yours sincerely,

Professor Ian Barr

Reviewer #1 (Remarks to the Author):

Stannard et al. reported characterization of a dual drug-resistant (NA-H275Y+PA-I38T) A(H1N1)pdm09 virus isolate and compared its replicative capacity, airborne transmissibility and relative fitness with those of the wild type and single mutant viruses (WT, NA-H275Y and PA-I38T), using both in vitro assays and a ferret model. It clearly demonstrated that this dual-mutant virus could efficiently transmit between ferrets via airborne exposure, but its viral fitness was moderately reduced compared with the WT virus. This is a carefully designed study with interesting, and quite informative results. I only have some minor comments:

1. Lines 16-18: As the two patients who shed viruses with treatment-emergent dual-mutations (NA-H275Y+PA-I38T) were previously reported in reference [30], in the Abstract, it would be better to clarify "Previously, two ... patients... were found ...", and "Here/In this study,". This can help the readers to distinguish the results from these two studies.
Comment accepted and manuscript amended.
2. Line 95-96: Whether the PA-P325Q may potentially have a role in facilitating the emergence of PA-I38T or stabilizing the mutant was not known or tested. It would be prudent to delete the statement "and therefore is unlikely to play a role in antiviral resistance or the stability of the PA-I38T mutation".
Comment accepted and manuscript amended.
3. Line 112, " high and low" should be "high or low".
Comment accepted and manuscript amended.
4. Line 152, "...WT or the..." should be "...WT and the..."
Comment accepted and manuscript amended.
5. Lines 243-244: actually, the ferret model used here could not distinguish the droplet and aerosol transmissions. It would be better to use the term "airborne transmission".
Comment accepted and manuscript amended. Aerosol changed to airborne in all iterations. As well as the following addition at line 492, “this cage design does not distinguish between airborne droplet and aerosol virus transmission.”

**WHO Collaborating Centre for Reference and Research on Influenza VIDRL
and Department of Microbiology and Immunology University of Melbourne**
Peter Doherty Institute for Infection and Immunity, Melbourne, VIC 3000, Australia
Postal Address: Locked Bag 815, Carlton South, VIC 3053, Australia
Tel +61 3 9342 9300 Fax + 61 3 9342 9329 Email ian.barr@influenzacentre.org
www.influenzacentre.org

6. Line 344, reference [30] (rather than [29]) should be cited.
Comment accepted and manuscript amended.
7. Full form of "PBS" should be spelled out the first time you use it in Line 353, rather than in Line 362.
Comment accepted and manuscript amended.
8. Lines 356, 447 and 448: "uL" should be "μL".
Comment accepted and manuscript amended.
9. Lines 374 and 457, "performed" should be "was performed" or "were performed" respectively.
Comment accepted and manuscript amended.
10. Line 381: "and GlaxoSmithKline, Australia, respectively": "and" and ", respectively" should be added to this sentence.
Comment accepted and manuscript amended.
11. Line 384: " ..., which was kindly provided by ...": "which" was missed in the sentence.
Comment accepted and manuscript amended.
12. Lines 402-403, a ") was missed after "(DMEM....TPCK-trypsin (SAFC Biosciences) ".
Comment accepted and manuscript amended.
13. Line 434, "Mustela putorius furo" should be italicized.
Comment accepted and manuscript amended.
14. Although many researchers use the term "post-infection", I think it would be more accurate to use "post-inoculation", as infections may not always immediately established after the "inoculation". Also, please be consistent in the use of this term throughout the main text, figures and supporting information.
Comment accepted and manuscript amended throughout.
15. Line 694, "nasal" should be "nasally" (i.e. "... were nasally washed...").
Comment accepted and manuscript amended.
16. Line 695, "... ferrets that were culled...": "that" should be deleted.
Comment accepted and manuscript amended.
17. Lines 696-697, should be " (b) Viral titre of each ... was enumerated...".
Comment accepted and manuscript amended.
18. Line 700, "statistical" should be "statistically".
Comment accepted and manuscript amended.
19. Line 738, " were WGS" should be "were sequenced using WGS" or "were whole genomically sequenced".
Comment accepted and manuscript amended.
20. For a phrase with extremely long attributive nouns, such as "patient day 1 post enrolment virus consensus genome (line 25, Supporting information)", it may be more readable to transform it into something like "virus consensus genome of the patient's day 1 post enrolment sample". Please go through the manuscript and see if you can reduce the use of such kind of long attributive structures.
Comment accepted and manuscript amended.
21. Line 47 of Supporting information: "... virus isolates were nasal (should be "nasally") washed..."; "Half of the ferrets that were culled": "that" should be deleted.
Comment accepted and manuscript amended.
22. Line 91 of Supporting information: "multiple comparisons test illustrate..." should be "multiple comparison tests illustrate..."; Line 93 "depicts" should be "depict"; Line 94 "comparisons" should be "comparison".
Comment accepted and manuscript amended.

23. Line 119 of Supporting information: "... (was not) nasal washed..." should be "...nasally washed...".
Comment accepted and manuscript amended.

Reviewer #2 (Remarks to the Author):

24. The authors have presented convincing data indicating that an A(H1N1)pdm09 variant with dual mutations (NA-H275Y and PA-I38T) resistant to both neuraminidase inhibitor and baloxavir marboxil had become less competent in infecting ferrets when co-administered with the WT virus, though the replicative fitness of the dual-mutant was similar to the WT virus in in vitro models. The findings are significant in terms of assessing the transmissibility of this dual mutant in community settings. The manuscript, however, focuses almost entirely on the description of the phenotypic observations of the double mutant without in-depth discussion on the possible mechanisms of the loss of viral fitness of the mutant in the ferret infection model with co-administered WT virus. Would it be the results of competition for the sialic acid receptors or efficiency in transcription of the viral genomes? Or would it be the differential amounts of defective viral particles generated by the mutant and the WT viruses? Or would it be the differential capability of the mutant and the WT viruses to escape host immune responses? Those mechanistic aspects are important to help the scientific community to gain further insights into the potential threat of these accumulating resistant mutations. The manuscript will be much improved if the authors may provide experimental results and speculations to explain the observed phenomenon.

Comment accepted and manuscript amended with the following at line 280-291; "Possible mechanisms that might explain the reduced competitive fitness of these mutant viruses include reduced enzymatic activity, changes in structural stability, or for the PA-I38T, increased genome replication errors resulting in additional defective interfering virus particles (DIVP). Enzymatic efficacy studies of the PA-I38T mutation have shown that this change reduced endonuclease enzymatic activity [15, 42], although polymerase complex activity was unchanged (or higher), compared to the WT virus polymerase [42, 50]. Computational structural analysis of NA-H275Y may indicate impaired stability compared to WT, although this virus also contains the NA mutations N369K and V241I, which were demonstrated by Hurt et al. to restore stability [53]. Our study did not find any differences in the proportion DIVP in the nasal wash samples from WT or NA-H275Y + PA-I38T infected ferrets as determined by WGS (looking for large deletions in individual gene segments). Hence, the true mechanism for the observed decrease in competitive fitness of this dual-mutant virus is likely to be multifaceted that may be difficult to fully elucidate."

Reviewer #3 (Remarks to the Author):

In this manuscript, the authors isolated influenza A/H1N1 2009 pandemic viruses with resistance to both neuraminidase inhibitors and baloxavir marboxil from immunocompromised patients and characterized the dual-resistant isolate in vitro and in vivo. Their results suggest that the dual-mutant influenza A/H1N1 2009 pandemic virus have a moderate loss of viral fitness compared to the wild-type virus.

The data presented in the manuscript are interesting and very important.

25. Fig. 2b. Did the authors confirm that the NA-H275Y and PA-I38T mutations were retained in the viruses recovered from the lungs of NA-H275Y+PA-I38T-infected animals on Day 5 post-infection?

Comment accepted and manuscript amended with the following at line 136-8, “We also observed by WGS that the NA-H275Y and PA-I38T mutations were retained in the virus samples recovered from ferret lungs, and no other amino acid changes were detected (data not shown).”

26. Page 3 line 96. “hemagglutinin (HA)-A204T”. The authors need to clarify whether this number means H1 or H3 numbering.

Comment accepted and manuscript amended with the following; “H1 numbering, beginning from Methionine.”

27. Page 4 line 112. “MOI” needs to be spelled out on the first use.

Comment accepted and manuscript amended.

REVIEWERS' COMMENTS:

Reviewer #3 (Remarks to the Author):

The authors responded appropriately to this reviewer's comments.